# Identification of Inflammation Markers as Novel Potential Predictors of the HIV-DNA Reservoir Size

**DOI:** 10.3390/ijms26178430

**Published:** 2025-08-29

**Authors:** Erick De La Torre Tarazona, Elisa Moraga, María Fons-Contreras, Raúl Vaquer, Sonsoles Sánchez-Palomino, Germán Vallejo-Palma, Sergio Calderón-Vicente, Sònia Vicens-Artés, Teresa Aldamiz-Echevarria, Marianela Ciudad Sañudo, Cristina Moreno, Inés Armenteros-Yeguas, Juan Tiraboschi, Sergio Reus Bañuls, José Alcamí, Sergio Serrano-Villar, Santiago Moreno

**Affiliations:** 1Infectious Diseases Department, Hospital Universitario Ramón y Cajal, Instituto Ramón y Cajal de Investigación Sanitaria (IRYCIS), 28034 Madrid, Spain; mafons20@gmail.com (M.F.-C.); raulvaquer@gmail.com (R.V.); secalder@ucm.es (S.C.-V.); serranovillar@gmail.com (S.S.-V.); 2Centro de Investigación Biomédica en Red de Enfermedades Infecciosas (CIBERINFEC), Instituto de Salud Carlos III, 28029 Madrid, Spain; ssanchez@recerca.clinic.cat (S.S.-P.); teresaldamiz@yahoo.es (T.A.-E.); cmoreno@externos.isciii.es (C.M.); 3HIV Unit, Infectious Diseases Service, Hospital Clinic, IDIBAPS, AIDS and HIV Research Group, University of Barcelona, 08036 Barcelona, Spain; moraga@recerca.clinic.cat (E.M.); svicens@recerca.clinic.cat (S.V.-A.); alcami@clinic.cat (J.A.); 4General Pediatrics and Infectious and Tropical Diseases Department, Hospital La Paz, 28046 Madrid, Spain; vallejopalma.g@gmail.com; 5Department of Biochemistry and Molecular Biology, Complutense de Madrid University, 28040 Madrid, Spain; 6Clinical Microbiology and Infectious Diseases Service, Hospital General Universitario Gregorio Marañón, Instituto de Investigación Sanitaria Gregorio Marañón, 28007 Madrid, Spain; 7Hospital Universitario de La Princesa, 28006 Madrid, Spain; ciudad13287@hotmail.com; 8Centro Nacional de Epidemiología, Instituto de Salud Carlos III, 28029 Madrid, Spain; 9Centro Sanitario Sandoval, Hospital Universitario Clínico San Carlos, Instituto de Investigación Sanitaria del Hospital Clínico San Carlos, 28010 Madrid, Spain; mariaines.armenteros@salud.madrid.org; 10HIV Unit and Infectious Diseases Service, Hospital Universitari de Bellvitge, 08907 Barcelona, Spain; jmtiraboschi@bellvitgehospital.cat; 11Infectious Diseases Unit, Hospital General Universitario Dr. Balmis de Alicante, Instituto de Investigación Sanitaria y Biomédica de Alicante, 03010 Alicante, Spain; reus_ser@gva.es; 12Department of Medicine, Alcalá University, 28871 Madrid, Spain

**Keywords:** HIV reservoir, intact proviruses, inflammatory markers, stem cell factor

## Abstract

The dynamics of the HIV reservoir during antiretroviral therapy (ART) exhibit variability, with a pronounced decline during the initial years of treatment. However, the identification of biomarkers and host factors associated with the decay of the different forms of HIV proviruses remains to be fully elucidated. We conducted a longitudinal study on people with HIV provided by the Spanish National HIV cohort. We assessed the HIV-DNA levels by Intact Proviral DNA Assay, and inflammatory markers using the Proximity Extension Assay, before and after ART initiation. A multivariate linear regression model was employed to identify potential predictive markers. Our results highlight the identification of novel inflammatory markers, such as ADA, DNER, CDCP1, SCF, among others, that varied significantly over ART initiation. In addition, we observed several markers associated with intact HIV-DNA before ART initiation (CD8A, CX3CL1, and ST1A1) or during undetectable viral load post-ART (IL-10). Moreover, up to five markers were able to predict the intact HIV reservoir decay over ART. The strongest predictor was Stem Cell Factor (SCF), where higher baseline levels of this marker were associated with a greater decline in the intact HIV reservoir. In conclusion, we have identified inflammatory markers associated with the size and dynamics of the HIV-DNA reservoir. These findings provide new insights that could contribute to the development of multi-targeted intervention strategies aimed at modulating or monitoring the HIV reservoir size.

## 1. Introduction

Despite the effectiveness of antiretroviral therapy (ART) in significantly reducing morbidity and mortality caused by Human Immunodeficiency Virus (HIV) infection, full eradication of the virus remains unattainable due to the presence of the HIV reservoir. Latently infected cells endure even under suppressive ART, leading to a viral rebound when therapy is discontinued [1,2]. Consequently, identifying the mechanisms involved in the dynamics of the HIV reservoir is an essential goal for the field.

The quantification of cell-associated HIV DNA is the gold standard for reservoir size measurement, although the classical methods employed usually tend to overestimate the reservoir size and cannot differentiate between replication-competent and non-intact genomes [3,4]. In contrast, the intact proviral DNA assay (IPDA) allows for greater throughput, resolution, and a more precise estimate of the frequency of replication-competent proviruses that most likely contribute to viral rebound. By using droplet digital PCR (ddPCR) to simultaneously target the packaging signal and env gene, the IPDA quantifies most of the intact and defective proviruses, allowing for a more accurate estimation of the HIV reservoir associated with the replication-competent reservoir [5,6].

The identification of biological markers associated with the HIV reservoir may be of great interest to better understand the biology of the virus and to design new therapeutic approaches. In this context, the search for cell markers that compose the reservoir has been attempted, identifying some markers, such as CD32, CD2, CD20, CD30, and PD-1, among others, that correlate with the size and dynamics of the viral reservoir [7]. Other works showed that specific epigenetic signatures may be correlated with reservoir dynamics and response to therapeutic interventions [8]. However, other studies employing single-cell proteogenomic profiling indicate that there is not a single unifying phenotypic marker, or even a combination of markers, that allows us to distinguish between infected and uninfected cells [9]. Thus, further research aimed to identify markers of the HIV-DNA reservoir could play a critical role in the search for a cure and in improving the design of individualized therapies.

On the other hand, despite ART, there is a consistent transcription of HIV RNA and translation of viral proteins, which may lead to chronic immune activation and persistent systemic inflammation [7]. In this context, HIV-infected individuals with high levels of viral replication in mucosal tissues exhibit increased concentrations of interleukin (IL)-6, tumor necrosis factor alpha (TNF-α), interferon (IFN)-γ, and IL-12 in the intestinal mucosa [10]. In addition, a longitudinal analysis showed that IL-1β, CXCL10, CXCL11, CXCL8, CCL2, and CCL4 remained elevated in the plasma of very-early-treated HIV-infected patients from a cohort of acute infection [11]. Moreover, persistent inflammation during ART may contribute to sustaining HIV persistence, as evidenced by the association between various plasma inflammatory cytokine levels with the establishment and maintenance of the HIV reservoir [12,13]. For instance, a recent study has identified Galectin-9 as a predictor of plasmatic cytokines of the intact HIV-DNA dynamics using a multiplex platform system [14].

However, it is essential to identify new markers of the size and/or dynamics of the HIV reservoir, and high-throughput technologies that allow for the evaluation of a high number of markers could contribute to this search. In addition, the interventions aimed to eradicate or reduce the HIV reservoir during the early phases of infection may contribute to obtaining a higher efficacy. For instance, the administration of drugs with latency-reversal properties during the viral phase before ART initiation could contribute to a stronger decline of the HIV-DNA reservoir [15,16]. Thus, the search for markers during this stage of the infection may contribute to developing personalized strategies for HIV elimination.

In this study, we aimed to identify inflammatory markers that could predict the size and changes in the HIV-DNA reservoir, particularly in the intact proviruses. Employing high-throughput technology, we found several inflammatory markers associated with the levels of the different forms of the HIV-DNA reservoir at pre-ART and post-ART phases. Moreover, our main finding is that baseline levels of inflammatory proteins, particularly Stem Cell Factor (SCF), were predictive markers of intact HIV kinetics over ART initiation. These findings provide new insights into potential mechanisms involved in the persistence and/or maintenance of the viral reservoir.

## 2. Results

### 2.1. Characteristics of the Patients

The clinical and sociodemographic characteristics of 23 PWH who were included in the study are summarized in Table 1. Briefly, before the ART initiation, the median viral load was 166,003 copies/mL (IQR: 19,657–853,933), the median age was 33 years (IQR: 25–40), 26% (6/23) of the individuals had a prior AIDS diagnosis, and only one participant (4%) had a confirmed hepatitis C virus (HCV) reported infection.

Regarding the initial treatment regimen, 22% (5) were on an integrase inhibitor-based regimen, 61% (14) on a non-nucleoside reverse transcription inhibitor-based regimen, and 17% (4) on a protease inhibitor-based regimen. Also, the median CD4 counts were 339 cells/μL (IQR: 91–575) and 600 cells/μL (IQR: 422–777) at pre-ART and post-ART, respectively. Most of the participants were male (91%), and the predominant mode of transmission was men who have sex with men (78%). The follow-up duration post-ART was 84 weeks (IQR: 72–110).

### 2.2. Levels of HIV-DNA Reservoir and Inflammatory Markers Drastically Reduce During ART Initiation

Proviral HIV-DNA levels, measured by IPDA, demonstrated a mean of 10-fold reduction following ART initiation (*p*-value < 0.0001). Total HIV-DNA levels declined from 7489 [IQR 1869–27020] to 857 [IQR 398–1829] copies per 10^6^ CD4+ T cells, representing an 8.7-fold reduction. Similarly, defective HIV-DNA levels decreased from 2585 [IQR 1089–18,260] to 649 [IQR 260–1739] copies per 10^6^ CD4+ T cells (4.0-fold reduction), while intact HIV-DNA levels showed the most pronounced decrease, from 1122 [IQR 527–5554] to 61 [IQR 14–191] copies per 10^6^ CD4+ T cells, representing an 18-fold reduction (Figure 1). The differential dynamics observed in the intact, total, and defective HIV reservoirs in our study are in agreement with previous studies that have reported comparable patterns, with a more pronounced decline observed in intact proviruses [6,17].

Moreover, up to 20 plasmatic inflammatory markers showed significant variations after the initiation of ART. Most of these proteins exhibited a decrease in their levels over ART, with the greatest changes observed in the C-X-C motif ligand: CXCL11 (3.7-fold), CXCL9 (4.8-fold), CXCL10 (2.7-fold), and interferon (IFN)-gamma (3.1-fold) (*p*-value < 0.01). The remaining markers, such as chemokine (C-C motif) ligand (CCL) 19, CCL3, Programmed Death-ligand 1 (PD-L1), adenosine aminohydrolase (ADA), CUB domain-containing protein 1 (CDCP1), among others, decreased within a range of 1.2- to 2.5-fold (*p*-value < 0.05). In contrast, stem cell factor (SCF) and Delta/Notch Like EGF Repeat Containing (DNER) proteins showed an increase in their levels post-ART, with fold changes of 1.5 and 1.2, respectively (*p*-value < 0.01) (Figure 2A,B, Appendix A). A heatmap and hierarchical clustering analysis with levels of these markers showed two main clusters, grouping most of the samples belonging to each pre-ART or post-ART time points (Figure 2C).

### 2.3. Identification of Inflammatory Markers Associated with the HIV Reservoir Size at Pre-ART and Post-ART Time Points

First, we attempted to identify levels of markers associated with the intact HIV-DNA size in different phases of the HIV infection. At pre-ART, during the detectable viral load phase, CD8 alpha (CD8A), C-X3-C motif ligand 1 (CX3CL1), and sulfotransferase 1A1 (ST1A1) markers were associated with the intact HIV-DNA size. Higher levels of CX3CL1 and CD8A were associated with higher levels of intact HIV-DNA, and higher levels of ST1A1 were associated with lower levels of intact proviral genomes (*p*-value < 0.05, Figure 3A). Moreover, the levels of the other six inflammatory markers (i.e., CXCL5, MMP-10, FGF-21) were associated with the size of the total and/or defective HIV-DNA reservoir (Appendix A).

Moreover, we also attempted to identify levels of markers associated with the intact HIV-DNA size at post-ART (during the undetectable VL phase). Higher levels of IL-10 were associated with lower levels of the intact HIV-DNA (Figure 3B). In addition, lower levels of CXCL9 and TNF were associated with a higher size of the defective HIV-DNA reservoir (Appendix A).

### 2.4. Baseline Level of Stem Cell Factor Is the Strongest Predictor of the Intact HIV-DNA Reservoir Decline

We also evaluated the influence of the levels and dynamics of the inflammatory markers on the HIV-DNA decline. First, we identified that the baseline (pre-ART) levels of five markers were able to predict the intact HIV-DNA decline over time. Baseline SCF (stem cell factor) levels were the strongest predictor marker, showing that a 2-fold increase in baseline SCF levels is associated with a 9.5-fold (10^0.98^) increase in the fold-change in the intact HIV-DNA reservoir. On the other hand, lower baseline levels of monocyte chemoattractant protein (MCP)-4, artemin (ARTN), CDCP1, and IL-8 proteins were associated with a higher intact HIV-DNA reservoir decline (Figure 4).

Additionally, baseline levels of matrix metalloproteinase (MMP)-10 were associated with the total and defective proviruses’ decline (Appendix A). Moreover, we observed that the variation in SCF and ARTN levels was also associated with the intact HIV-DNA reservoir dynamics over ART (Appendix A).

## 3. Discussion

The identification of new markers associated with the size of the HIV reservoir or able to predict its dynamics is relevant for assessing the effectiveness of therapeutic interventions aimed at reducing or eliminating the HIV-DNA. In this context, our study has proposed new potential predictors of HIV reservoir dynamics using a multiplex inflammatory markers platform.

The ART induces significant changes in the levels of various inflammatory markers in plasma, which is often observed as a part of the immune system’s response to the initiation of therapy, with certain markers either increasing or decreasing during the infection. Monitoring these marker profiles can provide valuable insights into the effectiveness of the treatment and the patient’s immune recovery or their association with HIV reservoir size. Among the cytokines most evaluated that are reduced over HIV treatment are tumor necrosis factor (TNF-α), Interleukin-6 (IL-6), IL-4, IL-10, transforming growth factor-beta (TGF-β), high-sensitivity C-reactive protein (hsCRP), soluble CD14 (sCD14), and sCD163, among others [18,19]. Our analyses showed several inflammatory markers decreasing over ART, such as CXCL11, CXCL9, CXCL10, IFN-gamma, CD8A, TRAIL, and IL10, among others (*q*-value < 0.15). The CXCL family of chemokines, including CXCL9, CXCL10, and CXCL11, plays a significant role in HIV infection. These molecules are involved in recruiting immune cells to sites of infection and inflammation. Elevated levels of CXCL9, CXCL10, and CXCL11 have been observed during primary HIV infection and are associated with disease progression [11,20]. Moreover, we also observed a reduction in some markers being minimally explored in the context of HIV infection, such as ADA, DNER, and CD244 (Figure 2). Functional analysis showed that these differentially expressed inflammatory markers are associated with different cell pathways, such as IL-6/JAK/STAT3 Signaling, Human Toll-like receptor signaling, Interferon Gamma Response, Chemokine-Mediated Signaling, and Cellular Response to Chemokine (*q*-value < 0.1, Appendix A). These findings indicate that these markers may play a role in the immune response to HIV and its treatment, and, consequently, their variation could indicate a shift towards immune reconstitution.

In the context of searching for associations with the size of the viral reservoir, particularly of the intact proviruses, we have identified some that could be of great interest (Figure 3). Studies have shown that during untreated HIV infection, there is an increase in the number and activity of CD8+ T cells as the immune system attempts to control viral replication. This heightened cytotoxic activity is a hallmark of the response to the virus prior to the initiation of ART [21]. In particular, CD8A is a component of the CD8 glycoprotein expressed on the surface of cytotoxic T lymphocytes, playing a crucial role in the immune response by acting as a co-receptor with the T-cell receptor (TCR). Its role is to recognize antigens presented by major histocompatibility complex class I molecules on infected cells, leading to the elimination of target cells. In the context of HIV infection before ART initiation, elevated levels of CD8A in plasma may indicate an increased activation of CD8+ T cell responses to the virus, which could be associated with the intact HIV reservoir. Moreover, we noted that ST1A1 levels are associated with the size of total, defective, and intact HIV-DNA. ST1A1 may play an important role in other viral infections [22]. In addition, a previous work employing the same multiplex platform of inflammatory markers reported that a cluster of proteins, including ST1A1, inversely correlated with the HIV reservoir size [23], which is consistent with our findings. On the other hand, CX3CL1 levels were also associated with the size of intact HIV-DNA in a prior report [11]. Interestingly, we also noted that CX3CL1 levels are higher in PWH with AIDS diagnosis before ART initiation (Appendix A). A functional analysis showed that with CD8, ST1A1 and CX3CL1 proteins are involved in several biological pathways related with immune response, such as lymphocyte-mediated immunity, negative regulation of interleukin-1 beta production, antigen receptor-mediated signaling pathway, negative regulation of macrophage activation, regulation of interleukin-1 alpha production, T cell-mediated immunity, and T cell receptor signaling pathway, among others (Appendix A, adjusted *p*-value < 0.15).

Regarding other predictive markers of the HIV reservoir size, we observed that levels of CXCL9 and CXCL5 were associated with the size of total and/or defective HIV-DNA, according to previous work that determined that higher levels of CXCL11 are associated with a higher HIV-DNA decline [13]. On the other hand, our results indicate that MMP-10 levels were associated with the size and dynamics of total and defective HIV-DNA. Prior reports indicate a dysregulation between MMP family proteins and their inhibitors, which contribute to the exit of HIV-infected cells from the bloodstream and the establishment of a reservoir in tissues [24]. On the other hand, IL-10 is an anti-inflammatory cytokine, and its increased levels in individuals with progressive HIV-1 disease have been observed. Consequently, blocking the IL-10 pathway may enhance the HIV-1-specific CD4+ T cell function [25]. Various studies have demonstrated that effective ART leads to a significant reduction in plasma IL-10 concentrations [19]. Interestingly, our results indicate that the levels of this cytokine at post-ART are inversely associated with the size of the intact HIV reservoir (Figure 3B). The limited identification of inflammatory markers associated with the size of the HIV reservoir during the undetectable VL phase is, at least in part, consistent with a recent report that found no inflammatory markers significantly associated with the viral reservoir in long-term, virally suppressed, treated individuals [26].

One of the findings we consider most relevant is the identification of SCF as a predictor of the dynamics of the intact HIV reservoir. This marker plays crucial biological roles in several processes, including hematopoiesis, which is essential for the development and survival of hematopoietic stem and progenitor cells, promoting their proliferation and differentiation into various blood cell types [27]. SCF contributes to immune system regulation, and its interaction with its receptor (c-KIT) can influence the survival, proliferation, and differentiation of different immune cells, including T cells [28]. Our results indicate that SCF increased over the course of ART, and their pre-ART levels were strongly associated with the reduction in the HIV reservoir (Figure 2, Appendix A). In the context of HIV and immune responses, SCF may contribute to the regulation of the CD4+ T cell number, which is crucial for immune function. However, the exact relationship between SCF levels and T cells may be complex and can vary depending on other factors, such as the stage of infection or ongoing treatment. On the other hand, we also observed that CXCL8 (IL-8) levels are associated with the HIV reservoir decay, which is in accordance with a prior report indicating the association of this marker with the HIV reservoir decline [11].

The primary aim of this study was to generate hypotheses and provide preliminary insights into novel inflammatory mechanisms that could be involved in HIV pathogenesis and the dynamics of the viral reservoir. However, this study is limited by a small sample size, due mainly to low sample availability of matching plasma and immune cells at the same time point. The exposed obstacles may restrict the generalization of the findings, identification of markers of HIV reservoir dynamics throughout long-term ART, thus masking the identification of other markers associated with the size or dynamics of the HIV reservoir, as well as the correction via the multiple comparison analysis. Additionally, we attempted to minimize the effect of potential confounding clinical parameters, adjusting the statistical multivariate models by baseline CD4 counts, which have been previously inversely associated with the HIV-DNA reservoir size [29,30].

In conclusion, our data show different markers that are associated with the HIV reservoir size or can predict their dynamics over ART, highlighting baseline pre-ART SCF levels as the strongest predictor of intact HIV reservoir decline. Further research to explore the role of these markers in the HIV pathophysiology is needed, since they may pave the way for personalized therapeutic approaches.

## 4. Materials and Methods

### 4.1. Patients and Samples

We recruited people with HIV (PWH) from the CoRIS Biobank (Madrid, Spain). The CoRIS is an open, prospective, multicenter cohort of confirmed HIV-infected adults, recruited since 2004 [31]. All participants provided written informed consent prior to enrollment in the study. Data follow the HIV Cohorts Data Exchange Protocol (HICDEP) with annual quality controls. We selected PWH who have available matched peripheral blood mononuclear cells (PBMCs) and plasma, at both pre-ART and post-ART time points. Stored PBMCs and plasma were provided by HIV Biobank.

The study was approved by the Research Ethics Committees of Instituto Ramón y Cajal de Investigación Sanitaria (20/070) and the CoRIS Review Board (RIS EPICLIN 07_2021), adhering to the Helsinki Declaration.

### 4.2. Determination of the HIV-DNA Reservoir Size

DNA was extracted from stored PBMCs using AllPrep DNA/RNA Mini Kit (Qiagen, Hilden, Germany), following the manufacturer’s instructions. We performed the Intact Provirus DNA Assay (IPDA), which consisted of two multiplex digital droplet (ddPCR) reactions performed using the QX600 Droplet Digital PCR System (Bio-Rad, Hercules, CA, USA): (a) one to distinguish between IPDA-intact and IPDA-defective proviruses (HIV-1 discrimination reaction) and (b) another to quantify DNA shearing and cell numbers (hRPP30 reaction) [5].

For HIV-1 proviral discrimination reactions, a mean of 1 µg of genomic DNA was analyzed in each reaction well. For DNA shearing and copy number reference reactions, 25 ng of genomic DNA was analyzed in each reaction well. Both reactions were adjusted using the DNA shearing index, which was maintained below 0.5 for all assays. The assay was conducted in six replicates, which were batch-processed and analyzed. The total HIV DNA (IPDA-total HIV-DNA) was calculated by summing intact and defective provirus copies. CEM.NKRCCR5 cells served as negative controls, while ACH-2 cells, containing a single HIV provirus per cell, were used as positive controls. Results were expressed as the number of HIV-DNA copies per million CD4+ cells.

### 4.3. Evaluation of Inflammation Marker Levels

We evaluate the levels of inflammatory markers in plasma at pre-ART and post-ART time points using a PEA (Proximity Extension Assay, OLINK, Uppsala, Sweden), which measures 92 inflammatory markers. The list of cytokines included in the Inflammation Panel is available on the OLINK website (https://olink.com/products/olink-target-96, accessed on 26 August 2025).

Plasma samples were pre-inactivated with Triton X-100 1% and stored at −80 °C prior to the evaluation of the inflammatory markers. The quantification of the markers was carried out in the COBIOMICS company (Córdoba, Spain). Results were expressed as Log2.

### 4.4. Statistical Analysis

Comparison of the HIV reservoir and inflammatory markers levels between pre-ART and post-ART was performed with a paired test. The correction by multiple comparison was performed with a false discovery rate (FDR) analysis, considering an *q*-value < 0.15 as statistically significant. To evaluate the association between the HIV reservoir size and inflammatory marker levels at different phases of HIV infection, we performed multivariate linear regression models adjusted by CD4 counts and inflammation marker levels (either at pre-ART or post-ART time points). To evaluate the impact of the marker levels on the kinetics of HIV-DNA over ART administration, we performed multivariate linear regression models considering the difference in the HIV reservoir between pre-ART and post-ART time points as the dependent variable and the other factors as the independent predictor variables. Statistical analysis and figures were realized with GraphPad Prism v8.

Functional analysis was carried out using two approaches. The Over-Representation Analysis (ORA), through the Enrichr API (https://maayanlab.cloud/Enrichr/, accessed on 26 August 2025), was used via the Python package gseapy (v1.1.6) to test gene sets using an Over-Representation Analysis (ORA). The protein lists (converted to gene symbols) significantly associated with conditions of interest in previous analyses were tested against three gene set databases: GO_Biological_Process_2025, MSigDB_Hallmark_2020, and KEGG_2021_Human. In addition, the Gene Set Enrichment Analysis (GSEA) was employed, using a ranking score computed by multiplying the t-test estimate by the negative logarithm (base 10) of the *p*-value, emphasizing proteins with strong effect sizes and statistical significance. This ranking was then Z-score-normalized to standardize the distribution. A GSEA-pre-ranked approach was applied using the Python package gseapy (v1.1.6). The ranked protein list was analyzed against the three gene set databases mentioned above. For both analyses, significant gene sets were selected based on an *q*-value < 0.15, and enrichment plots were generated for visualization of the significant results.

## Figures and Tables

**Figure 1 ijms-26-08430-f001:**
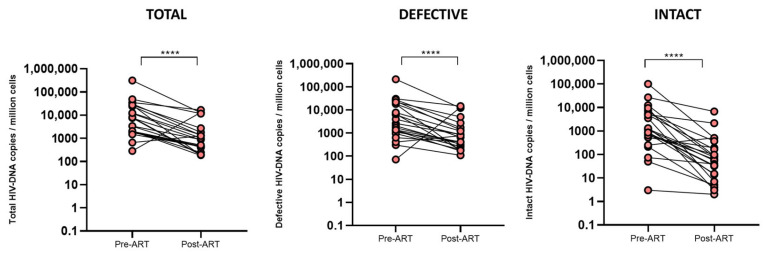
HIV-DNA reservoir reduction over ART administration. The HIV-DNA reservoir was determined in PBMCs, and values were normalized according to CD4 percentage data. Values below one copy of HIV-DNA per million cells were considered as 1. Statistical analysis was performed with the Wilcoxon test (**** *p*-value < 0.0001).

**Figure 2 ijms-26-08430-f002:**
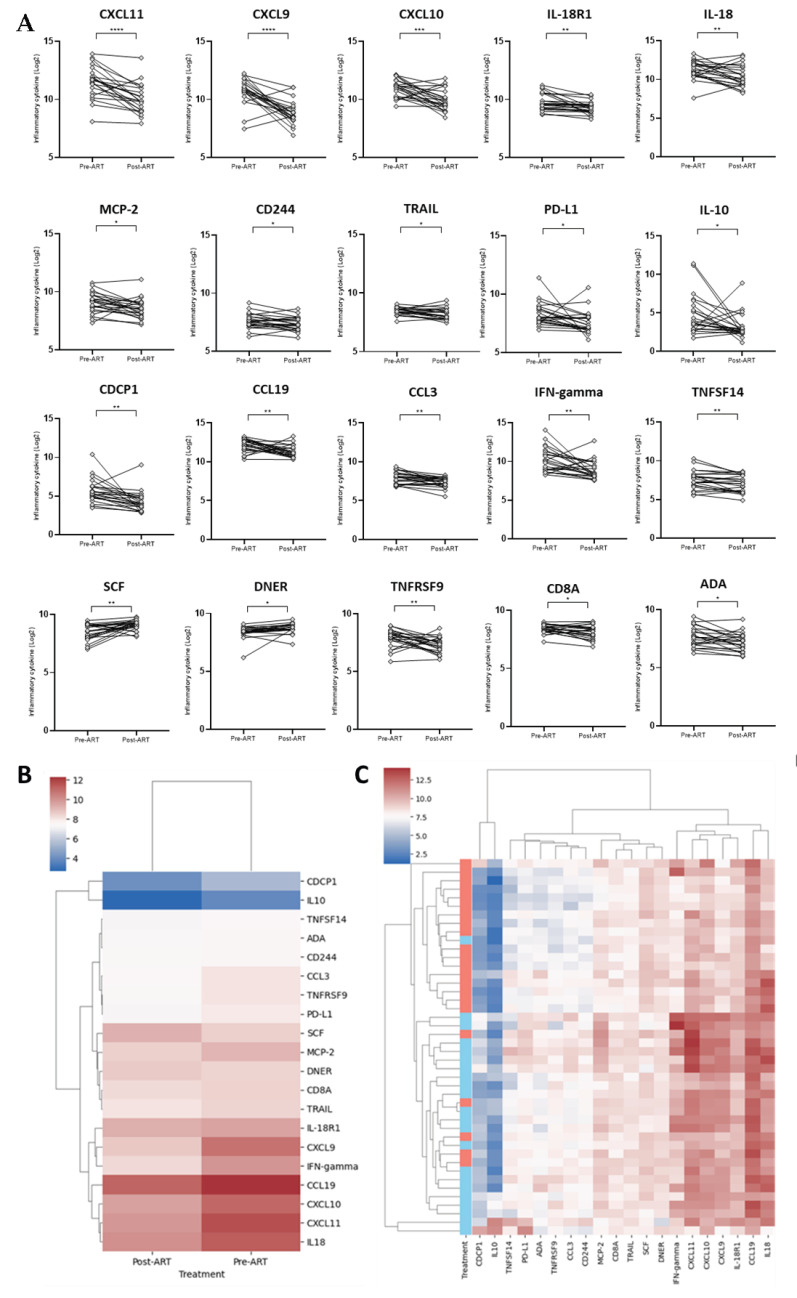
Variation in the inflammation markers over ART administration. (**A**) Differentially expressed markers among pre-ART and post-ART. Statistical analysis were performed with paired-test (* *p*-value < 0.05, ** *p*-value < 0.01, *** *p*-value < 0.001, **** *p*-value < 0.0001). All comparisons showed an *q*-value< 0.15 (Appendix A). (**B**) Heatmap showing the median levels of the differentially expressed cytokines among pre-ART and post-ART. (**C**) Heatmap with hierarchical clustering of the differentially expressed markers between pre-ART and post-ART. Treatment column: blue bars are samples from pre-ART, and red bars are samples from post-ART. Each row represents a participant.

**Figure 3 ijms-26-08430-f003:**
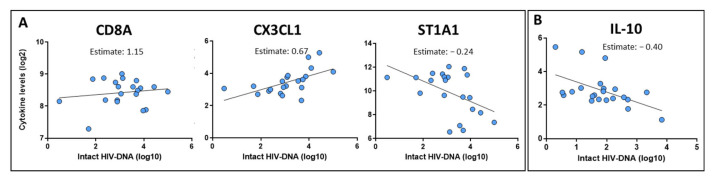
Inflammatory markers associated with the intact HIV reservoir size. Statistical analysis was performed using multivariate linear regression models adjusted by CD4 counts and inflammation marker levels at pre-ART (**A**) or post-ART (**B**). Levels of the intact HIV-DNA reservoir (Log10) at pre-ART (**A**) or post-ART (**B**) were considered as the dependent variable and the other factors as the independent variables.

**Figure 4 ijms-26-08430-f004:**
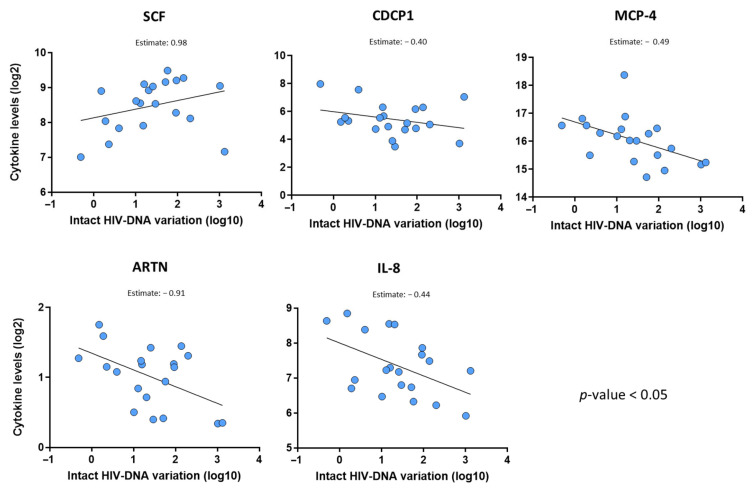
Inflammatory markers able to predict the intact HIV reservoir decline. Statistical analysis was performed using multivariate linear regression models adjusted by baseline (nadir) CD4 counts and inflammation marker levels at pre-ART. HIV-DNA reservoir (Log10) variation (difference between pre-ART and post-ART time points) was considered as the dependent variable, and the other factors as the independent predictor variables.

**Table 1 ijms-26-08430-t001:** Baseline clinical and socio-demographic parameters.

Baseline Clinical and Socio-Demographic Parameters	
**Number of individuals**	23
**Viral load** (HIV RNA copies/mL plasma)	166,003 (19,657–853,933)
**Pre-ART CD4 counts** (cells/mm^3^)	339 (91–575)
**Post-ART CD4 counts** (cells/mm^3^)	600 (422–777)
**Follow-up upon ART initiation** (weeks)	84 (72–110)
Age (years)	33 (25–40)
**AIDS diagnosis** Yes (n, %) No (n, %)	6 (26%) 17 (74%)
**HCV status** Yes No Unknown	1 (4%) 4 (17%) 18 (89%)
**Sex** Male (n, %) Female (n, %)	21 (91%) 2 (9%)
**Mode of transmission** MSM (n, %) Others (HSX, unknown) (n, %)	18 (78%) 5 (22%)
**Treatment regimen** INI-based (n, %) NNRTI-based (n, %) PI-based (n, %)	5 (22%) 14 (61%) 4 (17%)

CD4 counts, viral load, age, and follow-up time are expressed as median and IQRs (interquartile ranges). Viral load, age, HCV status, and AIDS status were considered before ART initiation. MSM: men who have sex with men; HSX: heterosexual. ART regimen was considered upon treatment initiation. NNRTI: non-nucleoside reverse transcription inhibitors. INTI: integrase inhibitors. PI: protease inhibitors. All ART regimens include two nucleoside reverse transcription inhibitors.

## Data Availability

The data that support the findings of this study are available from the corresponding author upon reasonable request.

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
