# Peer review of "Identification of Inflammation Markers as Novel Potential Predictors of the HIV-DNA Reservoir Size"

_ijms, 2025, doi:10.3390/ijms26178430_

Round 1

Reviewer 1 Report

Comments and Suggestions for Authors

The current manuscript by Tarazona et al., titled “Identification of inflammation markers as novel potential predictors of the HIV-DNA reservoir size” describes a longitudinal study on people with HIV provided by the Spanish National HIV cohort. Here they assessed the HIV-DNA levels by Intact Proviral DNA Assay, and inflammatory markers by Proximity Extension Assay, before and after ART initiation.  They find that total HIV-DNA levels declined 8.7-fold, defective HIV-DNA levels decreased by 4.0-fold; and intact HIV-DNA decrease 18-fold. When looking at the cytokine data, they saw a decrease in the C-X-C motif ligand CXCL 11 (CXCL11) (3.7-fold), CXCL9 (4.8-fold), CXCL10 (2.7-fold), and interferon (IFN)-gamma (3.1-fold) post ART. Main finding was that baseline levels of Stem Cell Factor (SCF) were the strongest predictive marker of intact HIV kinetics over ART. Overall, I found the manuscript interesting and important, however, I have few comments:

  1. Why is there a decrease in the defective HIV copy number post ART (fig. 1)?
  2. Data in Fig. 2A is interesting, but what is the connection to ART and intact vs. defective DNA?
  3. Was similar number of cells used for the data in figs.3 and 4?
  4. Is the data on Stem Cell Factor correlated with NK or B-cell activity in these patients?

Author Response

We express our gratitude to the Reviewer 1 for the time and evaluation on our manuscript and for the positive feedback. We are happy to comment on your remarks below. Comments of the referee are in blue, followed by our response in black.

The current manuscript by Tarazona et al., titled “Identification of inflammation markers as novel potential predictors of the HIV-DNA reservoir size” describes a longitudinal study on people with HIV provided by the Spanish National HIV cohort. Here they assessed the HIV-DNA levels by Intact Proviral DNA Assay, and inflammatory markers by Proximity Extension Assay, before and after ART initiation.  They find that total HIV-DNA levels declined 8.7-fold, defective HIV-DNA levels decreased by 4.0-fold; and intact HIV-DNA decrease 18-fold. When looking at the cytokine data, they saw a decrease in the C-X-C motif ligand CXCL 11 (CXCL11) (3.7-fold), CXCL9 (4.8-fold), CXCL10 (2.7-fold), and interferon (IFN)-gamma (3.1-fold) post ART. Main finding was that baseline levels of Stem Cell Factor (SCF) were the strongest predictive marker of intact HIV kinetics over ART. Overall, I found the manuscript interesting and important, however, I have few comments:

  1. Why is there a decrease in the defective HIV copy number post ART (fig. 1)?

The smaller reduction observed in the defective HIV reservoir compared to the intact reservoir following ART initiation, could be explained by the lower immune clearance pressure exerted on defective proviruses [Bruner et al., Cell 2016;].

Some prior reports indicate a similar pattern of the HIV reservoir decline, showing also a higher decline of intact proviruses in comparison to defective ones over ART. We have added a phrase in results section (Line 160) indicating that our observation is according to preliminary data:

“The differential dynamics observed in the intact, total, and defective HIV reservoirs in our study are in agreement with previous studies that have reported comparable pat-terns, with a more pronounced decline observed in intact proviruses [17,18]”

2. Data in Fig. 2A is interesting, but what is the connection to ART and intact vs. defective DNA?

This figure aims to highlight the inflammatory markers in our panel that change following to ART. These results allowed us to confirm the variation of several previously reported markers, such as CXCL9, CXCL10, and TNF, among others, supporting the robustness of the method in detecting known changes associated with HIV infection. In addition, our analysis revealed changes in other proteins following ART initiation that had not been previously associated with HIV infection, suggesting potential novel biomarkers or mechanisms involved in treatment response. The association of these inflammatory markers with the size and dynamics of the intact and defective HIV reservoir is shown in the following figures or supplementary tables.

3. Was similar number of cells used for the data in figs.3 and 4?

As indicated in the Methods section, immune cells were used to determine the HIV reservoir. A similar number of cells was used for each assay, adjusted according to DNA content (an average of 1 µg per reaction, depending on availability). In each individual, inflammatory marker levels were also quantified by PEA in paired plasma samples at each time point. Thus, in Figures 3 and 4, the number of cells employed for HIV reservoir quantification are the same.

4. Is the data on Stem Cell Factor correlated with NK or B-cell activity in these patients?

Unfortunately, we did not measure specific markers of NK or B cell activity in our study due to samples limitations, and therefore we are unable to determine whether there is a correlation between these responses and SCF levels.

We would like to thank the referee again for the relevant comments and taking the time to review our manuscript.

Reviewer 2 Report

Comments and Suggestions for Authors

The authors' topic is novel in that it evaluates novel potential predictors of HIV-DNA reservoir size. The authors also conduct a comprehensive literature review. The findings with the measurement of 92 inflammatory markers in plasma at pre-ART and post-ART are significant; however, I have some recommendations: 1. The markers will be different (at least in CD8A, CX3CL1, and ST1A1m, IL-10, and SCF) depending on whether the patient has an AIDS diagnosis or has a high viral load. 2. The authors emphasize the identification of Stem Cell Factor (SCF), but it is not as significant as the differences in the concentration of inflammatory cytokines were greater in others such as CSCL11, CXCL9, and CXCL10.

Author Response

Reviewer 2

We would like to thank to Reviewer 2 for the time to review our manuscript and for pointing out some aspects to improve our work. We have gone through all the comments and we addressed each of them below. Comments of the referee are in blue, followed by our response in black.

The authors' topic is novel in that it evaluates novel potential predictors of HIV-DNA reservoir size. The authors also conduct a comprehensive literature review. The findings with the measurement of 92 inflammatory markers in plasma at pre-ART and post-ART are significant; however, I have some recommendations:

1. The markers will be different (at least in CD8A, CX3CL1, and ST1A1m, IL-10, and SCF) depending on whether the patient has an AIDS diagnosis or has a high viral load.

We thank to the reviewer for this important comment. When comparing pre-ART levels of the inflammatory markers SCF, CD8, ST1A1 and CX3CL1 across groups stratified by low (<250 cells/mm3) or high CD4 counts (>250 cells/mm3), low (<100,000 copies HIV-RNA) or high viral load (>100,000 copies HIV-RNA), and AIDS diagnosis, we did not observe any statistically significant differences in the most of these comparisons. We only noted higher levels of CX3CL1 in PWH with AIDS diagnosis before ART initiation, and this observation will be mentioned in the discussion section. The following table shows these results, and will be added at the supplementary material of the manuscript:

CD4 counts

Viral load

AIDS diagnosis

Low CD4

High CD4

p-value

Low VL

High VL

p-value

Yes

No

p-value

CD8

8.5

[7.9 - 8.6]

8.4

[8.2 - 8.9]

0.25

8.3

[8.2 – 8.9]

8.5

[8.2 – 8.6]

0.50

8.3

[8.1 – 8.7]

8.5

[8.2 – 8.8]

0.74

ST1A1A

9.5

[8.0 - 11.4]

10.9

[8.9 - 11.2]

0.49

10.8

[9.5 -11.3]

9.5

[7.2 - 11.3]

0.21

10.2

[8.4 – 11.0]

10.9

[7.8 -11.4]

0.98

CX3CL1

3.7

[3.0 - 4.5]

3.1

[2.8 - 3.7]

0.82

2.9

[2.7 – 4.4]

3.5

[3.1 – 4.0]

0.63

3.9

[3.1 – 6.2]

3.1

[2.7 – 3.8]

0.03

SCF

8.2

[7.7 - 8.7]

8.9

[8.1 - 9.1]

0.14

8.6

[8.0 – 9.1]

8.5

[7.9 – 9.1]

0.88

8.7

[8.0 – 9.3]

8.5

[8.0 – 9.1]

0.42

* Values are indicated as medians and IQR. Statistics: comparative analysis were realized with t-test

It is possible that no differences are found in the most of these comparisons due that the identification of these markers associated with the HIV reservoir size or decay was carried out with multivariate linear regression models, which enables the assessment of the independent effect of each variable while controlling for potential confounders. This approach can reveal differences and associations that may remain undetected in univariate analyses.

2. The authors emphasize the identification of Stem Cell Factor (SCF), but it is not as significant as the differences in the concentration of inflammatory cytokines were greater in others such as CSCL11, CXCL9, and CXCL10.

We thank the reviewer for this comment. As the reviewer correctly notes, the variation in CXCL9, CXCL10, and CXCL11 levels (decrease) is greater than that observed for SCF (increase) following ART administration. However, we emphasized the observation regarding SCF because its baseline levels are associated with the magnitude of the reduction in the intact proviruses. Therefore, SCF may be considered a potential predictive marker of the replication-competent HIV reservoir decay in PWH.

Finally, we would like to thank again the referee for taking the time and effort to review our manuscript.